# Decomposing wealth-based inequalities in ideal cardiovascular health in Kenya

James Odhiambo Oguta [1] ✉, Penny Breeze[1], Catherine Akoth[1], Elvis Wambiya[1], Grace Mbuthia[2], Peter Otieno[3], Gladwell Gathecha[4], Elizabeth Onyango[4], Yvette Kisaka[4] & Peter J. Dodd [1]

## Abstract

**Background** This study examined wealth-related inequalities in ideal cardiovascular health (iCVH), as defined by the 2010 American Heart Association guideline, among Kenyan adults. **Methods** The study analysed data from 3816 adults who participated in the 2015 World Health Organization (WHO) STEPwise survey on non-communicable disease risk factors. The concentration index (C) and concentration curves were used to quantify inequalities in overall iCVH and its seven-component metrics, and a Wagstaff-type decomposition analysis was performed to identify the main factors contributing to the observed inequalities. **Results** A pro-rich inequality (higher prevalence in individuals with wealth) is observed in overall iCVH (C = 0.08; $p = 0.006$), which is more pronounced among women. Pro-rich inequalities are also evident for ideal body mass index (C = 0.31; p < 0.001), ideal blood pressure (C = 0.16; p < 0.001), and ideal total cholesterol (C = 0.15; p = 0.005). Conversely, pro-poor inequalities (higher prevalence in individuals living in poverty) are observed in ideal nicotine exposure (C = −0.10; p = 0.012) and fruit and vegetable intake (C = −0.09; p = 0.048). No significant inequalities are detected for ideal fasting blood glucose (C = −0.03; p = 0.534) or physical activity (C = 0.05; p = 0.291). Decomposition analysis shows that urban residence (31.4%), wealth (30.7%), region (16.5%), and education (8.5%) contribute most to the observed pro-rich inequality in iCVH. **Conclusions** Socioeconomic inequalities for iCVH in Kenya are more prevalent in people with wealth, particularly among women. Addressing these disparities requires equity-oriented, gender-sensitive prevention policies targeting people living in poverty and less-educated populations, especially in urban settings.

## Plain english summary

This study examines how wealth affects heart health in Kenya. Using national survey data from 2015, we measure wealth disparities in "ideal cardiovascular health," a set of seven behaviours and health factors linked to the risk of heart disease. We find that wealthier people are more likely to have ideal heart health, especially among women. People with wealth have healthier body weight, blood pressure, and cholesterol levels, while people living in poverty individuals are less likely to smoke and more likely to eat fruits and vegetables. Factors such as living in urban areas, income, education, and region of residence explain most of the inequality. To reduce these gaps, heart-health programmes should focus on people living in poverty, less-educated groups, particularly women living in urban areas.

Cardiovascular diseases (CVD) are a major public health concern globally and in Sub-Saharan Africa (SSA), and are the leading causes of non-communicable disease (NCD) morbidity and mortality[1]. Like many other countries in the SSA region, Kenya has undergone an epidemiological and demographic transition during the last three decades leading to the rise in CVD burden, with stroke and ischemic heart disease currently being the sixth and ninth leading causes of deaths and disability[2]. In 2021, the five main risk factors for CVD (high blood pressure (BP), dietary risks, high fasting blood glucose (FBG), high body mass index (BMI) and tobacco) were among the top 10 drivers of all-cause deaths and disability in Kenya[2].

In Kenya, 24.5% of the adult population live with hypertension[3], 13.5% use tobacco[4], 18.3% consume a high salt diet, while only 6% consume the

minimum required fruit and vegetable (FV) servings[5]. Moreover, about 85.8% of Kenyan adults have at least four NCD risk factors[6]. Tackling the high burden of CVD risk factors in Kenya requires scaling up both primordial and primary CVD prevention interventions. To support a comprehensive assessment of cardiovascular health (CVH) status in the population, the American Heart Association (AHA), in 2010, introduced the concept of "Ideal Cardiovascular Health (iCVH)"[7]. The 2010 AHA criteria for assessing iCVH uses combined scores across seven CVH metrics including four health behaviours (diet, tobacco intake, physical activity and BMI) and three biological factors (BP, FBG and total cholesterol). The use of multiple CVD risk factors to assess an individual's CVH status is more specific than using individual risk factors, whose absence may not imply

[1]Sheffield Centre for Health and Related Research, Division of Population Health, School of Medicine and Population Health, University of Sheffield, Sheffield, UK. [2]School of Nursing, College of Health Sciences, Jomo Kenyatta University of Agriculture and Technology, Nairobi, Kenya. [3]African Population and Health Research Center, Nairobi, Kenya. [4]Division of Cancer and Non-Communicable Diseases, Ministry of Health, Nairobi, Kenya. ✉e-mail: mcogutajamo@gmail.com

ideal CVH. Previous studies have established that possessing iCVH metrics reduces the risk of CVD (coronary heart disease and stroke), and all cause and CVD-related mortality[8–11].

Addressing inequalities in the distribution of various CVH metrics is crucial for halting and reversing the rising trend in CVD burden in Kenya. While previous studies in SSA have only assessed the prevalence and factors associated with iCVH[10,12–18], none has examined the socioeconomic inequalities in ideal CVH using composite CVH metric. Moreover, no previous SSA study has applied decomposition analysis to examine the determinants of the socioeconomic inequalities in iCVH. In Kenya, less than half of the adult population have an ideal CVH, while only 1.2% have the maximum possible CVH score[18]. Previous studies examining socio-economic inequalities in individual CVD risk factors in Kenya found a higher concentration of the burden of hypertension[19] and tobacco use[20] among the people living in poverty. However, there is a lack of evidence on the extent of socioeconomic inequalities in iCVH and its component metrics in Kenya, as well as differences among males and females.

Understanding the magnitude of the socioeconomic inequalities and their determinants could help to inform the design and implementation of equity-focused interventions targeting different groups to improve the overall CVH status of the Kenyan population. Therefore, this study examines the magnitude of wealth-based inequalities in iCVH and its component metrics in Kenya, and explores the drivers of the observed inequalities as well as their differences among males and females.

This study reveals marked wealth-based disparities in iCVH in Kenya. A significant pro-rich inequality is observed in overall iCVH, particularly among women, with the greatest inequalities seen in body mass index, blood pressure, and total cholesterol. Conversely, people living in poverty are more likely to exhibit ideal nicotine exposure and fruit and vegetable intake. Decomposition analysis indicates that urban residence, wealth status, region, and education are the main contributors to the observed inequality. The study findings highlight the importance of addressing socioeconomic and structural determinants to promote equitable cardiovascular health outcomes in Kenya.

## Methods
### Study setting
Kenya is a lower middle income country located in East Africa, with a population of about 54 million and per capita Gross Domestic Product of 1980 United States dollars (USD) in 2023[21]. Since 2013, Kenya introduced a devolved system of government composed of one national government and 47 counties, which are semi-autonomous. The national government is responsible for formulating health policy and management of national referral health facilities, while counties oversee primary health care and health service delivery from the community level to county referral health facilities.

### Data sources and study population
This study used data from the 2015 World Health Organization (WHO) STEPwise (STEPS) survey for NCD risk factors in Kenya[22]. The STEPS survey was the first and latest available nationally representative surveillance of different NCD risk factors conducted between April and June 2015, involving adults aged 18–69 years. All the 47 counties were included in the study.

### Sampling and sample size
Multistage cluster sampling was used to select a representative sample of adults drawn from the fifth National Sample Surveys Evaluation Programme (NASSEP V) master sampling frame maintained by the Kenya National Bureau of Statistics (KNBS). Data were collected from 4500 individuals using standard WHO questionnaires. Additional details of the STEPS survey and its procedures are reported elsewhere[22].

### Measures
**Outcome variable.** Ideal CVH was defined by the cut-offs proposed by the AHA 2010 guideline based on history of CVD, four health behaviours (smoking, physical activity, diet, BMI) and three health factors (blood pressure, glucose, and cholesterol levels)[7,23]. Although a more recent 2022 AHA guideline is available[24], this study used the 2010 guideline as it provides explicit cutoffs for defining ideal levels of each CVH metric, facilitating standardised measurement and comparability. Table 1 presents the operational definition of the seven CVH metrics used to construct the outcome variable. Each CVH metric was defined as "ideal", "intermediate" or "poor" using the cutoff stipulated by the AHA 2010 guideline[7]. We used the reported daily fruit and vegetable intake as a proxy for the diet metric due to the absence of structured data on other dietary components.

We defined iCVH as an individual with ideal levels of at least five out of the seven CVH metrics in the absence of an existing CVD. To generate the seven-point iCVH metric, we dichotomised each CVH metric by assigning a score of 1 if it met the criteria for ideal and 0 if not ideal (poor and inter-mediate). The scores for each individual risk factor were then summed up to generate an ordinal scale of between 0 and 7, with a score of 7 indicating that all the ICH metrics were at ideal levels. We classified the seven-point score as 0–2 (poor CVH), 3–4 (intermediate CVH) and 5–7 (ideal CVH). Individuals with CVH scores of at least five were then assigned a score of 1 (ideal CVH) and those with 0–4 assigned zero (non-ideal CVH). We performed sensitivity analysis using poor CVH status (0–2 iCVH metric) as an

## Table 1 | Operational definition of ideal cardiovascular health

| Metric | Poor | Intermediate | Ideal |
|---|---|---|---|
| Body mass index | ≥30 kg/m² | 25–29.9 kg/m² | Below 25 kg/m² |
| Smoking | Currently smoking or using smokeless tobacco | Past smoker or user of smokeless tobacco ≤12 months | Has never smoked or quit >12 months |
| Fruit and vegetable intake | 1 serving or none | 2–4 servings | Five or more servings |
| Physical activity ≥ | <600 metabolic equivalent of task (MET) minutes per week | 600−1500 metabolic equivalent of task (MET) minutes per week | >1500 metabolic equivalent of task (MET) minutes per week |
| Blood pressure (BP) | Systolic BP (SBP) of ≥140 mmHg or diastolic BP (DBP) of ≥90 mmHg or existing hypertension patient | SBP 120 ≤ 139 mmHg or DBP 80 ≤ 89 mmHg | SBP < 120 mmHg or DBP < 80 mmHg without antihypertensives |
| Blood Sugar | Fasting blood glucose (FBG) ≥ 7 mmol/l (≥126 mg/dl) or existing diabetic patient | FBG of 5.6–6.9 mmol/l (100–125 mg/dl) | FBG of <5.6 mmol/l (<100 mg/dl) without treatment |
| Total cholesterol (TC) | TC of ≥6.3 mmol/l (≥240 mg/dl) | TC of 5.2–6.2 mmol/l (200–239 mg/dl) | <5.2 mmol/l (200 mg/dl) |
| CVH Status | 0–2 ideal metrics | 3–4 ideal metrics | 5–7 ideal metrics |

*CVH* Cardiovascular Health, *TC* Total Cholesterol, *FBG* Fasting blood glucose, *MET* Metabolic Equivalent of Task, *SBP* Systolic Blood Pressure, *DBP* Diastolic Blood Pressure.

outcome variable, which was defined as individuals who had only two or less CVH metrics at ideal levels or those with previous history of CVD based on self-reported history of myocardial infarction (MI), heart attack or stroke.

**Measure of household socioeconomic/wealth status.** Principal component analysis (PCA) was used to construct the wealth index, which was the measure of the household socioeconomic position based on data from household assets[25,26]. PCA examines the variation in a linear combination of household asset variables to generate an index that can be used as a proxy measure of household wealth status[26]. The household assets/characteristics included in the PCA were type of toilet facility, type of house (floor, walling and roofing materials), cooking fuel, source of drinking water, availability of electricity, ownership of electronics (radio, television, refrigerator, washing machine, phones, computers), assets for transportation (bicycle, motorcycle, car or truck, animal drawn cart), livestock ownership, ownership of agricultural land and employment of a house help. The first principal component from the linear combination in continuous scale (wealth score) was converted into five wealth quintiles[26], which include poorest (first), poorer (second), middle (third), richer (fourth) and richest (fifth).

**Other explanatory variables.** Other independent variables included in this study were age (18–29 years, 30–39 years, 40–49 years and 50–69 years), sex (female and male), marital status(in a union and not in a union), place of residence (rural and urban), highest level of education completed (no formal education, primary education, and secondary-and-above), alcohol intake (current user and non-user) and region (Rift Valley, Nyanza, Central, Eastern, Western, North Eastern, Coast and Nairobi). Ethnicity was defined as Kalenjin, Kamba, Kikuyu, Luhya, Maasai, Borana, Kamba, Embu, Meru, Somali, Mijikenda, Luo, Turkana, Kisii and others.

### Statistics and reproducibility

We used STATA 18.5 (Stata Corporation, College Station, Texas) to perform the statistical analyses and R Statistical Software (Version 4.4.2) to perform multiple imputation. All the statistical analyses were adjusted for the clustered sampling design by applying the *svyset* command using the 200 selected clusters as the primary sampling units, rural-urban residence as strata and assigned sampling weights. Frequencies and percentages were used to describe the sample characteristics and prevalence of iCVH by socio-demographic characteristics. We used Pearson's Chi-Square test of Independence to assess the association between iCVH and other covariates. Statistical significance was evaluated at $p < 0.05$. Data were presented using tables and figures. The code used in this study can be accessed from this Github repository[27].

### Assessing the wealth-based inequalities in iCVH.

To assess the wealth-based inequalities in iCVH and its metrics, we first plotted the concentration curve (CC) of the cumulative proportion of the population ranked by iCVH status against the cumulative proportion of the population ranked by their wealth status[26,28,29]. The CC is interpreted with regards to its position relative to the line of equality (y = x). Where the CC lies above (below) the line of equality, the health variable (iCVH) is considered to be more concentrated among the poor (rich)[26]. We also computed the concentration index (C), which is a summary measure of the socioeconomic inequality in iCVH. The C is defined as twice the area between the concentration curve and the line of equality[26]. Mathematically, the C is defined as[26]:

$$C = \frac{2}{n\mu} \sum_{i=1}^{n} h_i R_i - 1$$

where; $h_i$ is the health (iCVH) variable in the $i$th individual; μ is the mean iCVH; $n$ is the number of people, while $R_i$ is the fractional rank of the $i$th individual by their socioeconomic status from the poorest to the richest.

The C ranges between −1 and +1, with negative values indicating that the health variable (iCVH) is more concentrated among the poor (pro-poor), while positive C indicates a pro-rich distribution[26,30,31]. In this study, 'pro-rich' inequality refers to a positive concentration index, indicating that ideal cardiovascular health (or the specific CVH metric) is more prevalent among people with wealth. Conversely, 'pro-poor' inequality refers to a negative concentration index, indicating higher prevalence among individuals living in poverty. For binary outcome variables like iCVH, however, the C is not usually bounded between −1 and +1 but depends on the μ. Wagstaff's normalisation technique was applied to correct the bounds of the C using the following formula[26]:

$$C_{\text{normalised}} = \frac{C}{1 - \mu}$$

The delta method was used to compute the standard errors using the *nlcom* command in Stata. We computed the C for each of the seven CVH metrics alongside the overall iCVH.

### Examining the drivers of the observed inequalities.

Wagstaff-type decomposition analysis was applied to examine the drivers of the observed wealth-based inequalities in iCVH[26]. Decomposition analysis represents the overall C as the proportion contributed by individual factors and takes into account the sensitivity of health to factors (elasticity) and its income-related inequality (measured by $C_k$)[26]. Starting with a linear additive model of a continuous health variable (y);

$$y = \alpha + \sum_{k} \beta_k x_k + \varepsilon$$

the corresponding C is rewritten as;

$$C = \sum_{k} (\beta_k \underline{x}_k / \mu) C_k + GC_\varepsilon / \mu \qquad \text{where;}$$

μ is the mean of y; $\underline{x}_k$ is the mean of each explanatory variable, $x_k$; $C_k$ is the C for each predictor, $x_k$, whereas $GC_\varepsilon$ is the generalised C for the error term (ε).

The contribution of each explanatory variable ($x_k$) is thus a product of the term $\beta_k \underline{x}_k / \mu$ (elasticity of y with respect to $x_k$) and its concentration index ($C_k$). When aggregated, the contributions of individual $x_k$ constitute the explained part of the C, with the last term of the equation ($GC_\varepsilon / \mu$) being the residual part.

Since iCVH is a dichotomised outcome variable, we performed Wagstaff normalisation to compute the C and then used a probit model to compute the marginal effects of the covariates ($\beta_k$), which were then used to compute the contributions of the individual determinants to the C[26]. We stratified our analyses by sex by computing the C to assess the inequalities in iCVH among males and females.

### Handling missing data.

We performed complete case analysis, involving the exclusion of all missing observations, whose results are reported in the main text of this paper. As a sensitivity analysis, we imputed missing data by multiple imputation using chained equations (MICE) using the MICE package in R[32,33]. We first assessed the pattern of missingness, which revealed that data were missing at random, justifying the use of MICE for imputation. Predictive mean matching was used to impute continuous data, logistic regression used for binary data, while polytomous logistic regression was used to impute unordered categorical data[32]. We first performed 20 imputations, which were increased to 80 imputations after performing convergence diagnostics. All imputation diagnostics are presented in Supplementary Figs. 1–6. The imputed results were consistent with those from the complete case analyses (Supplementary Tables 1–4).

### Ethics statement

The protocol for the 2015 STEPS survey was reviewed and approved by the Scientific and Ethics Review Unit (SERU) of the Kenya Medical Research Institute (KEMRI). All study procedures were conducted in accordance with the principles of the Declaration of Helsinki[22]. Written informed consents were obtained from all participants prior to data collection. For this secondary analysis, the protocol and study tools were reviewed and approved by the Moi Teaching and Referral Hospital (MTRH)- Moi University Institutional Research and Ethics Committee (approval number 00046773) and the ethics committee at the University of Sheffield, Division of Population Health. Moreover, a research permit was granted by the National Commission for Science, Technology and Innovation (NACOSTI) in Kenya (License Number- NACOSTI/P/24/33160).

## Results

### Sample characteristics

After excluding 684 participants with missing observations in the complete case analysis, the final sample included 3816 adults aged 18–69 years. A higher proportion of the sample were females, aged below 30 years, in a marital union, had no formal education, were self-employed or unemployed, non-drinkers of alcohol, and resided in rural areas and the Rift Valley region. About one third of the sample belonged to the Kalenjin and Kikuyu ethnic groups (Table 2).

### Prevalence of ideal cardiovascular health metrics

Table 2 presents the overall prevalence of iCVH by sample characteristics. The overall prevalence of iCVH (5–7 metrics) was 51.9% (95% CI; 49.1–54.6%). The iCVH prevalence was slightly higher in males than females (53.7% vs 50.0%). The prevalence decreased with increasing age but was higher among the unmarried (56.3%) and those residing in rural areas (55.7%) compared to the married (49.5%) and urban residents (45.5%), respectively. The Nyanza region had the highest iCVH prevalence (60.0%), while the Central region reported the lowest (40.5%). These patterns were consistent in the sex-stratified analyses (Table 2).

Figure 1 presents the prevalence of ideal levels of individual CVH metrics. The highest proportion of the population had ideal FBG (90.6%) and cholesterol levels (90.5%), while ideal FV intake (6.3%) and blood pressure (29.9%) had the lowest prevalence. Males had significantly higher prevalence of ideal cholesterol (92.8% vs 88.2%, $p < 0.01$), FBG (92.5% vs 88.7%, $p < 0.05$) and BMI (83.3% vs 61.2%, $p < 0.001$) metrics while females had significantly higher prevalence of ideal nicotine exposure (94.1% vs 63.7%, $p < 0.001$) and blood pressure (33.1% vs 26.8%, $p < 0.01$) metrics. There were no sex differentials in the prevalence of ideal FV intake and physical activity metrics (Fig. 1).

### Distribution of iCVH metrics by wealth quintile

Figures 2 and 3 present the prevalence of iCVH metrics by wealth quintile. The prevalence of overall ideal CVH metric was highest among the richer and richest quintiles among all adults and females but not males (Fig. 2).

Among all adults, there were statistically significant differences in the prevalence of ideal BMI ($p < 0.001$), blood pressure ($p < 0.001$) and cholesterol ($p = 0.032$) metrics by quintile, with the highest prevalence among the richest and richer quintiles. Similarly, the prevalence of overall iCVH was highest among the richest quintiles but not statistically significant ($p = 0.094$). Statistically significant differences were observed in the prevalence of ideal CVH, smoking, BMI and BP metrics by quintile among females. Among males, we observed statistically significant wealth-based disparities in the prevalence of ideal physical activity ($p = 0.008$), BMI ($p < 0.001$), BP ($p < 0.001$) and cholesterol levels ($p = 0.02$) by wealth quintile (Fig. 3).

### Wealth-based inequalities in ideal CVH

The overall C was positive (C = 0.08; 95% CI: 0.02, 0.14; $p = 0.006$), indicating that iCVH was more concentrated among the rich (pro-rich). Similar pro-rich inequalities in iCVH were observed in the sex-stratified analyses

although only females (C = 0.11; 95% CI: 0.03, 0.18; $p < 0.01$) had a statistically significant C (Supplementary Table 1).

Figure 4 presents the concentration curves (CC) for the whole sample and by sex. The CC for all adults and females were below the line of equality, indicating significant concentration of the overall iCVH among rich households. However, the CC for males crossed the line of equality indicating that the observed wealth-based inequality is not statistically significant.

### Wealth-based inequalities in individual iCVH metrics

Figure 5 and Supplementary Table 1 present the results of the C for the wealth-based inequalities in iCVH metrics. Statistically significant pro-rich inequalities were observed in ideal BMI and BP metrics in the combined sample, and among males and females. There were statistically significant pro-rich inequalities in ideal total cholesterol levels in the combined sample and among males. Ideal physical activity among males were more concentrated among the people with wealth, whereas the converse was true among females who had a negative C (although not statistically significant). There were significant pro-poor inequalities in ideal FV intake in the combined sample and ideal smoking metric among all the population groups (Fig. 5 and Supplementary Table 1). Supplementary Table 2 presents the results of the concentration indices for the wealth-based inequalities in poor CVH metrics, which had consistent patterns with inequalities in ideal CVH metrics. Similarly, the results from the imputed analyses (Supplementary Tables 3 and 4) were consistent with those of complete case analyses.

### Explaining the drivers of the observed wealth-based inequality in iCVH

Figure 6 and Supplementary Table 5 present the results of the decomposition analysis for the contribution of individual factors to the observed wealth-based inequality in iCVH. Urban residence (31.4%) and wealth status (30.7%) were the highest contributors to the observed inequality in iCVH among all adults. Region of residence (16.4%) was the third leading contributor, mainly attributed to Nairobi (6.1%) and Central (5.0%) regions, followed by respondent's education level (8.5%). Alcohol intake and ethnicity (led by the Kikuyu ethnic group at 2.4%) contributed 5.4% and 5.3% of the observed inequality, respectively. Age and employment status only contributed 1.7% and 0.6% of the observed inequality, respectively (Fig. 6).

Almost half (45.2%) of the observed inequality in iCVH among females was explained by household wealth (Fig. 6). This was followed by urban residence (24.4%), education level (12.8%), region of residence (9.8%, with Nairobi contributing 7.6%) and ethnicity (7.1%, with Kikuyu contributing 2.1%). Among males, urban residence (27.6%), wealth index (27.2%), and region (24.6%) were the leading contributors to the observed inequality followed by ethnicity (9.9%) and occupation (6.2%). Age and alcohol intake contributed only 1.9% and 2.5% of the observed inequality in iCVH among females, respectively (Fig. 6).

The variables included in the model explained 72.9%, 76.4% and 77.7% of the observed inequality in iCVH among all adults (residual of 0.023 representing 27.1%), females (residual of 0.025 representing 23.6%) and males (residual of 0.017 representing 22.3%), respectively (Supplementary Table 5).

## Discussion

The present study found higher prevalence of ideal CVH, BMI, BP and cholesterol metrics among richer and richest wealth quintiles. Moreover, statistically significant pro-rich inequalities, less prevalent in people living in poverty, were observed in the distribution of ideal CVH among all adults and females. Similar pro-rich inequalities were observed in ideal cholesterol and physical activity metrics among males. On the contrary, pro-poor inequalities were observed in the distribution of the ideal smoking metric in all the study groups and ideal FV intake in the combined study sample. The results of the sensitivity analyses performed with poor CVH metrics as outcomes were consistent with those of iCVH metrics. Household wealth

**Table 2 | Sample characteristics and prevalence of ideal cardiovascular health by sample characteristics**

| | Sample | Prevalence of ideal cardiovascular health (5–7 CVH metrics) | | | |
| --- | --- | --- | --- | --- | --- |
| | | Overall | | Female | Males |
| Variable | n (%) | n | % (95% CI) | n = 2248 | n = 1568 |
| Total | 3816 (100) | 1838 | 51.9 [49.1,54.6] | 50.0 [47.2, 52.8] | 53.7 [49.6, 57.6] |
| **Sex** | | | | | |
| Female | 2248 (58.9) | 1076 | 50.0 [47.2, 52.8] | - | - |
| Male | 1568 (41.1) | 762 | 53.7 [49.6, 57.6] | - | - |
| **Age group (years)** | | | | | |
| <30 | 1242 (32.6) | 777 | 63.5 [59.4, 67.4] | 59.5 [54.2, 64.6] | 67.5 [62.9, 71.7] |
| 30–39 | 1065 (27.9) | 528 | 50.2 [45.1, 55.4] | 49.2 [43.1, 55.4] | 51.3 [43.7, 58.8] |
| 40–49 | 697 (18.3) | 278 | 39.9 [34.2, 46.0] | 43.1 [35.9, 50.6] | 37.4 [29.6, 45.9] |
| 50+ | 812 (21.3) | 255 | 31.6 [26.9, 36.6] | 30.0 [24.4, 36.3] | 33.3 [26.6, 40.8] |
| **Marital Status** | | | | | |
| Married | 2590 (67.9) | 1222 | 49.5 [46.0, 53.0] | 50.6 [46.9, 54.2] | 48.2 [42.5, 54.1] |
| Single | 1226 (32.1) | 616 | 56.3 [53.2, 59.3] | 48.8 [44.1, 53.5] | 62.3 [56.9, 67.4] |
| **Education** | | | | | |
| No formal | 1523 (39.9) | 743 | 52.4 [48.3, 56.5] | 51.1 [46.9, 55.2] | 54.3 [47.3, 61.0] |
| Primary | 1236 (32.4) | 595 | 52.6 [48.0, 57.1] | 50.1 [44.8, 55.4] | 55.2 [47.3, 62.9] |
| Secondary + | 1057 (27.7) | 500 | 50.5 [44.7, 56.2] | 48.1 [42.6, 53.7] | 51.9 [43.0, 60.8] |
| **Occupation** | | | | | |
| Employed/Salaried | 715 (18.7) | 314 | 46.7 [39.3, 54.3] | 46.8 [38.4, 55.4] | 46.7 [37.7, 56.0] |
| Self-Employed | 1536 (40.3) | 704 | 52.1 [48.4, 55.8] | 48.6 [43.5, 53.7] | 54.8 [50.2, 59.3] |
| Unemployed/Unpaid | 1565 (41) | 820 | 54.3 [50.8, 57.8] | 51.6 [47.1, 56.1] | 59.7 [53.6, 65.6] |
| **Wealth quintile** | | | | | |
| Poorest | 644 (16.9) | 240 | 44.6 [36.4, 53.2] | 47.0 [39.4, 54.8] | 43.0 [31.6, 55.2] |
| Poorer | 736 (19.3) | 343 | 51.7 [46.5, 56.8] | 42.1 [35.0, 49.4] | 59.8 [50.5, 68.4] |
| Middle | 793 (20.8) | 401 | 51.7 [46.8, 56.6] | 46.5 [41.1, 52.0] | 58.3 [51.7, 64.5] |
| Richer | 814 (21.3) | 417 | 55.6 [50.7, 60.4] | 57.4 [52.1, 62.6] | 53.9 [45.9, 61.6] |
| Richest | 829 (21.7) | 437 | 55.6 [50.3, 60.8] | 55.7 [50.1, 61.2] | 55.5 [47.5, 63.3] |
| **Alcohol intake** | | | | | |
| Never/Past drinker | 3022 (79.2) | 1534 | 55.6 [52.9, 58.2] | 51.1 [48.1, 54.1] | 62.1 [57.4, 66.7] |
| Current user | 794 (20.8) | 304 | 40.5 [34.3, 46.9] | 39.4 [30.2, 49.3] | 40.7 [33.7, 48.2] |
| **Residence** | | | | | |
| Rural | 1986 (52) | 1046 | 55.7 [52.3, 58.7] | 53.9 [50.4, 57.3] | 57.6 [53.1, 61.8] |
| Urban | 1830 (48) | 792 | 45.5 [41.6, 49.4] | 42.6 [39.2, 46.1] | 47.8 [41.0, 54.8] |
| **Region** | | | | | |
| Rift Valley | 1188 (31.1) | 613 | 55.8 [51.6, 59.9] | 52.6 [47.3, 57.9] | 58.8 [54.2, 63.2] |
| Eastern | 695 (18.2) | 303 | 50.2 [43.6, 56.5] | 51.2 [45.4, 56.8] | 49.2 [39.6, 58.7] |
| Nyanza | 487 (12.8) | 267 | 60.0 [51.6, 67.8] | 55.6 [47.9, 63.1] | 64.8 [54.8, 73.7] |
| Coast | 419 (11) | 177 | 44.6 [36.0, 53.6] | 44.7 [34.9, 54.9] | 44.5 [33.5, 56.1] |
| Nairobi | 51 (1.3) | 21 | 47.2 [42.6, 51.7] | 39.1 [32.6, 45.9] | 54.0 [47.2, 60.8] |
| Western | 367 (9.6) | 186 | 56.4 [50.6, 61.9] | 53.3 [44.1, 62.3] | 58.9 [51.1, 66.3] |
| North Eastern | 177 (4.6) | 90 | 58.0 [46.0, 69.1] | 51.0 [40.7, 61.8] | 69.9 [53.4, 82.5] |
| Central | 432 (11.3) | 181 | 40.5 [31.4, 50.4] | 46.9 [31.4, 62.8] | 34.1 [21.5, 49.4] |
| **Ethnicity** | | | | | |
| Kalenjin | 644 (16.9) | 349 | 56.5 [51.1, 61.8] | 54.3 [45.6, 62.8] | 58.3 [49.1, 67.6] |
| Embu | 86 (2.3) | 36 | 47.1 [30.3, 64.6] | 45.0 [21.9, 70.5] | 49.9 [30.5, 69.5] |
| Kikuyu | 612 (16) | 249 | 46.7 [41.8, 51.8] | 45.9 [39.8, 52.1] | 47.5 [39.1, 56.1] |
| Kamba | 347 (9.1) | 151 | 47.6 [41.7, 53.6] | 51.5 [44.8, 58.3] | 40.7 [30.4, 51.9] |
| Borana | 18 (0.5) | 7 | 34.8 [17.8, 56.7] | 48.3 [20.3, 77.4] | 24.7 [8.6, 53.2] |
| Kisii | 195 (5.1) | 107 | 56.3 [44.2, 67.7] | 60.3 [46.7, 72.6] | 52.9 [36.5, 68.8] |
| Luhya | 475 (12.5) | 228 | 51.7 [46.2, 57.1] | 47.3 [38.8, 55.7] | 55.6 [49.1, 61.9] |

**Table 2 (continued) | Sample characteristics and prevalence of ideal cardiovascular health by sample characteristics**

| | Sample | Prevalence of ideal cardiovascular health (5–7 CVH metrics) | | | |
|---|---|---|---|---|---|
| | | Overall | | Female | Males |
| Luo | 414 (10.9) | 217 | 54.4 [44.9, 63.5] | 45.9 [36.9, 55.1] | 64.0 [52.6, 73.9] |
| Maasai | 59 (1.6) | 34 | 58.5 [37.4, 76.9] | 58.9 [34.6, 79.5] | 58.1 [34.4, 77.7] |
| Meru | 221 (5.8) | 87 | 48.4 [35.0, 62.0] | 46.1 [30.7, 62.0] | 49.7 [34.6, 64.8] |
| Mijikenda | 143 (3.8) | 59 | 42.5 [31.9, 53.8] | 47.3 [35.5, 59.5] | 36.6 [20.8, 56.0] |
| Somali | 186 (4.9) | 96 | 56.4 [44.6, 67.6] | 49.8 [39.4, 60.6] | 67.3 [51.5, 79.9] |
| Turkana | 82 (2.2) | 40 | 54.2 [41.0, 67.6] | 45.3 [37.2, 53.7] | 70.8 [53.6, 83.6] |
| Other | 334 (8.8) | 178 | 57.8 [47.2, 67.8] | 63.0 [52.4, 72.5] | 54.0 [40.6, 66.9] |

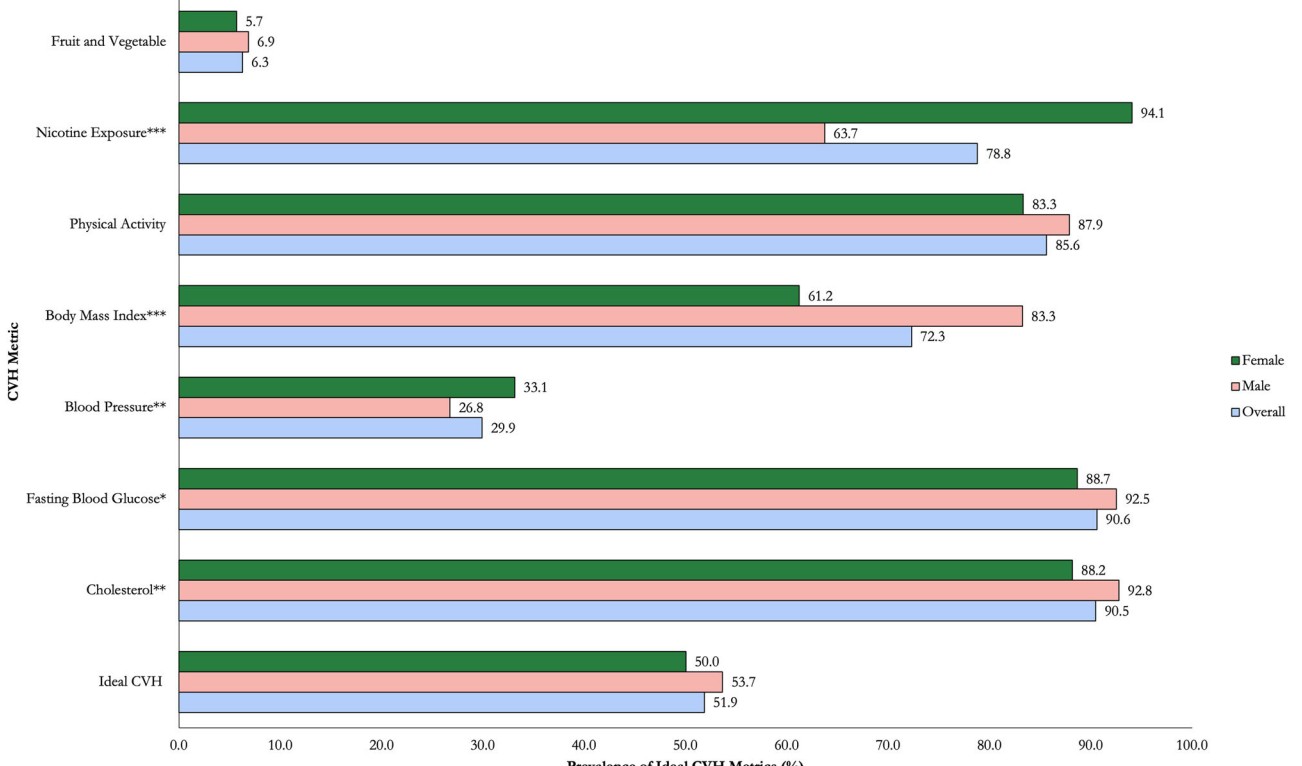

**Fig. 1 | Prevalence of ideal levels of individual cardiovascular health metrics by sex.** A graph presenting the weighted prevalence of ideal cardiovascular metrics by sex (male and female) and among all adults (overall). The total sample was 3816 adults (2248 females and 1568 males). Green colour represents females; pink represents males; Blue colour represents the combined sample (all adults); CVH Cardiovascular Health; All the p-values were two sided; *** - Statistically significant (Chi2 test) at $p < 0.001$; ** - Statistically significant (Chi2 test) at $p < 0.01$; * - Statistically significant (Chi2 test) at $p < 0.05$.

status, urban residence, region of residence and respondent's education level were the highest contributors to the observed socioeconomic inequalities in iCVH across all the study groups. Nairobi and Central regions had the highest contribution to the proportion of the explained component of the observed socioeconomic inequalities attributed to the region of residence. Other factors that explained the observed inequalities included ethnicity (mainly driven by the Kikuyu ethnicity), occupation (mainly among males), alcohol intake and age. Education level contributed about one tenth of the observed inequality among all adults (8.5%) and females (12.8%), but was not an important contributor to the observed inequality among males.

To the best of our knowledge, this is the first study to assess wealth-based inequalities in iCVH in SSA. Previous studies applying the AHA definition of iCVH within the SSA setting have only focused on estimating its prevalence and determinants[10,12–17], with none examining the extent of wealth-based inequalities in CVH status at the population level. A few studies have examined wealth-based inequalities in the distribution of individual CVD risk factors including hypertension[19,34], obesity[35–37], diabetes[38], and tobacco use[20,39]. Of all the global studies that examined inequalities in multiple CVD risk factors[40–47], only one study in Tunisia[46] assessed the inequalities in high cardiovascular risk, which was a composite score defined by the presence of at least three (out of five) CVD risk factors (hypertension, diabetes, hypercholesterolaemia, obesity, and tobacco use). Our study thus contributes to the understanding of the magnitude of inequalities in iCVH in SSA to inform the design of relevant interventions to address the rising burden of CVDs in the region.

Our finding of pro-rich concentration of iCVH contrasts with those of a Tunisian study, which reported a higher concentration of high CVD risk (defined by at least three out of five CVD risk factors) among individuals from households with high socioeconomic status[46]. Also, the findings are contrary to those of previous studies in Botswana[41], South Africa[35,37] and using pooled data from multiple countries[36,44] that reported higher concentration of overweight and obesity among richer households. The pro-

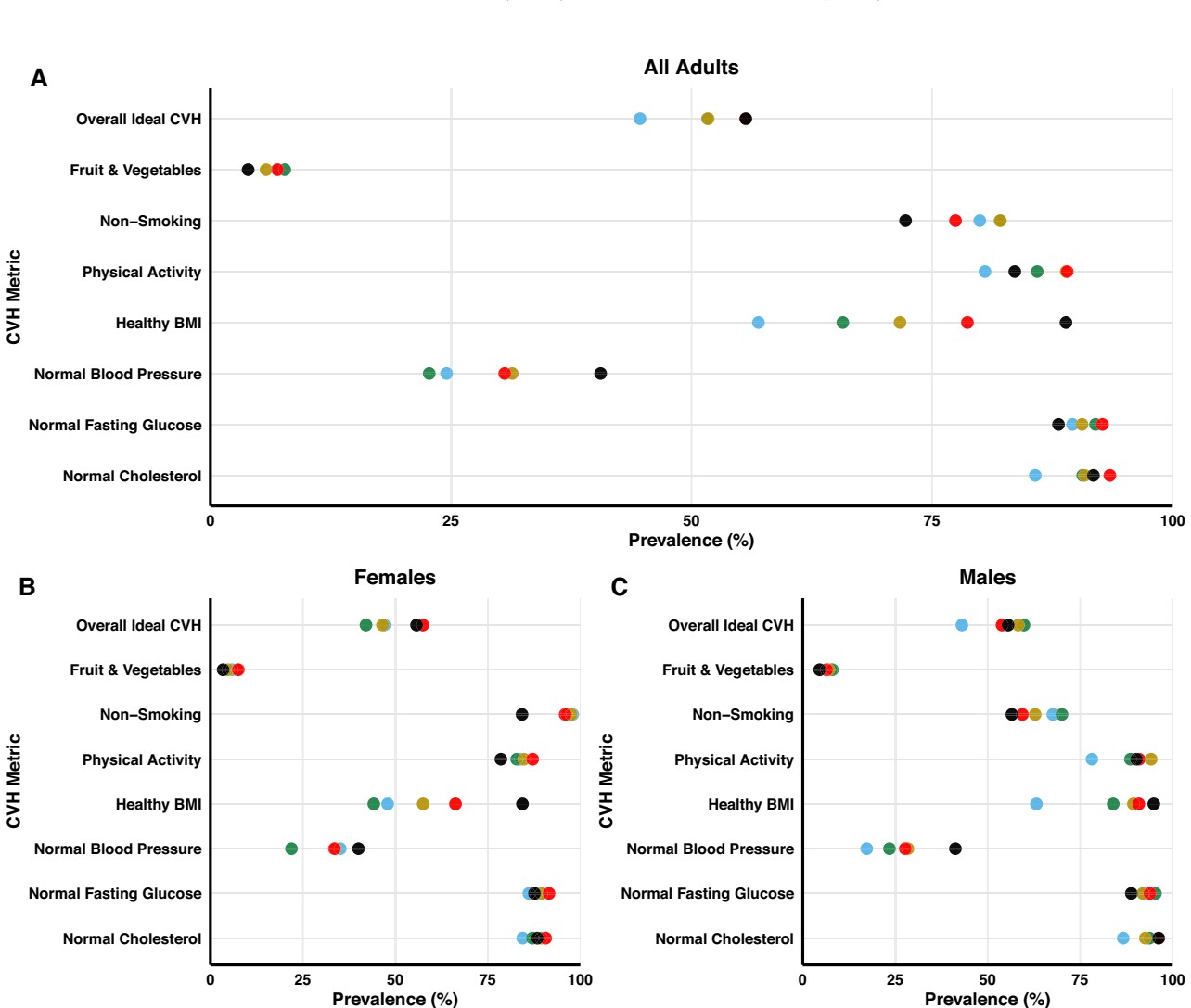

**Fig. 2 | Equiplots for the prevalence of ideal cardiovascular health metrics by wealth quintile.** **A** An equiplot graph presenting the weighted prevalence of ideal cardiovascular health metrics by wealth quintile among all adults (*n* = 3816). **B** An equiplot graph presenting the weighted prevalence of ideal cardiovascular health metrics by wealth quintile among females (*n* = 2248). **C** An equiplot graph presenting the weighted prevalence of ideal cardiovascular health metrics by wealth quintile among all adults (*n* = 1568). **A–C** Blue coloured dots represent the weighted prevalence for the poorest (first) wealth quintile; Green coloured dots represent the weighted prevalence for the poorer (second) wealth quintile; Yellow coloured dots represent the weighted prevalence for the middle (third) wealth quintile; Red coloured dots represent the weighted prevalence for the richer (fourth) wealth quintile; Black coloured dots represent the weighted prevalence for the richest (fifth) wealth quintile; CVH Cardiovascular Health.

rich inequality in ideal BP (positive C) reported in our study agrees with the results of a Kenyan study[19] that reported significant concentration of hypertension among people living in poverty, but is contrary to those of a study using pooled data from women across 33 SSA countries[44]. Previous studies have also reported higher concentrations of low physical activity among the poor in Zambia[45], but among the rich in Zimbabwe[45] and Botswana[41]. The study finding on pro-rich concentration of ideal cholesterol agrees with those of another study in rural Uganda[43], which found low HDL cholesterol levels to be more common among individuals from low socio-economic status. On the contrary, we found significant pro-poor concentration of ideal nicotine exposure metric in all study groups and ideal FV intake in the combined sample. Similarly, poor nicotine exposure metric and poor FV intake were significantly concentrated among people with wealth. Previous studies reported significant concentrations of tobacco use[20,40–42,44,45] and inadequate FV consumption[40,41,45] among people living in poverty. However, a study in Namibia found that cigarette smoking was more

concentrated among the people with wealth[39], which agrees with our study findings. This complex picture of regional variation at the component level of CVH highlights the importance of locally specific data and analyses, relevant to different contexts.

Despite the progress made towards reducing inequalities over the last few decades, Kenya still has an income Gini index of 0.485, which indicates persistently high levels of income inequalities characterised by stark regional and urban-rural disparities[48]. Furthermore, access to all types of preventive and promotive health care interventions in Kenya is more available for people with wealth[49]. For instance, while the need for hypertension screening in Kenya is higher among people living in poverty, access to hypertension screening and treatment is more concentrated among people with wealth[50]. The observed pro-rich inequality in iCVH in Kenya highlights the need to bridge the gap between people with wealth and people living in poverty with regards to improving the distribution of health and wellbeing. Relevant health promotion and CVD prevention interventions targeted at

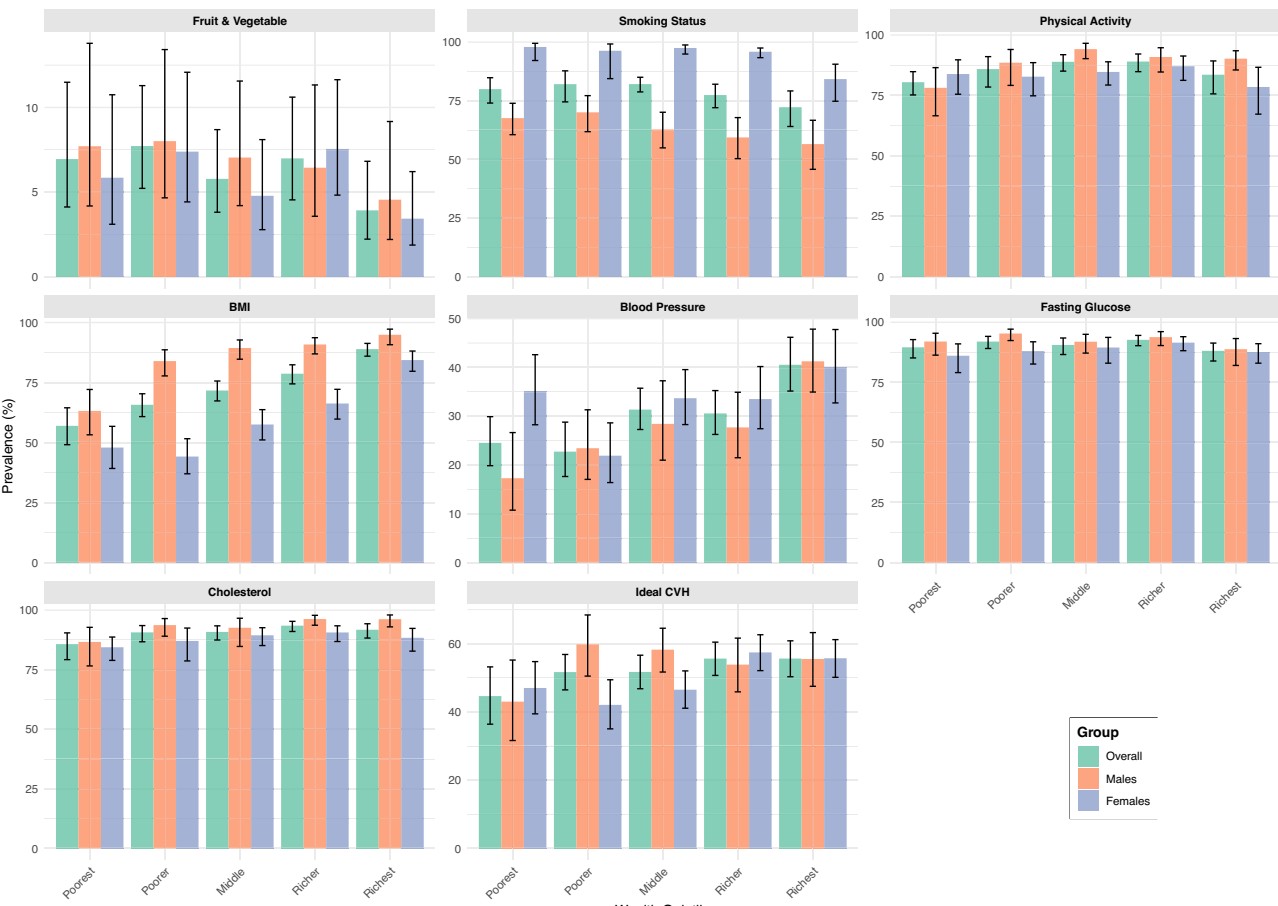

**Fig. 3 | Prevalence of ideal cardiovascular health metrics by wealth quintile.** A graph presenting the weighted prevalence of individual ideal cardiovascular health metrics by wealth quintile among all adults (overall) and by sex (males and females). The total sample was 3816 adults (2248 females and 1568 males). Error bars represent the 95% Confidence Interval: CVH- Cardiovascular Health.

the people living in poverty are required to address the observed disparities in the overall distribution of iCVH in Kenya.

While the statistically significant pro-rich inequality in overall iCVH is small in magnitude (C = 0.08), this masks larger inequalities for constituent CVH metrics, and among males and females. The highest pro-rich inequalities were found in ideal BMI metric, with a C of 0.33 ($p < 0.001$) in females, 0.45 ($p < 0.001$) in males and 0.31 ($p < 0.001$) for the combined sample. Similar patterns were observed in poor BMI metric (obesity), which was consistently concentrated among poor households. These findings point at the significant burden that poor BMI exerts upon people living in poverty in Kenya necessitating the design of targeted interventions to address high BMI in this group.

We observed variations in the magnitude of socioeconomic inequalities in iCVH metrics by sex. Females had higher magnitude of inequality in the overall iCVH metric and smoking metrics, while males had higher magnitude of inequality in the BMI, BP and total cholesterol metrics. Whereas males had a statistically significant pro-rich inequality in ideal physical activity metric, females had a pro-poor inequality, though not statistically significant. A previous study that used pooled data for women from 33 SSA countries found pro-rich inequalities in high BP (C: 0.1352), overweight/obesity (C: 0.2285), and tobacco use (C: -0.2551)[44]. The sex differences in iCVH observed in this study may partly stem from inequalities in domestic and occupational activities, health-seeking behaviour, and sex-specific lifestyle risk factors such as physical activity and diet. Women in SSA tend to seek healthcare services more frequently than men, which may contribute to higher detection rates of cardiometabolic conditions such as hypertension and type 2 diabetes[17,51,52]. Our results suggest the need for policy makers and practitioners to adopt a gender lens while designing and

implementing interventions aimed at improving the CVH status of the population.

Urban residence was the leading contributor (31.4%) to the observed inequalities in iCVH in Kenya. Previous studies in Kenya have found increased CVD risk in urban areas in Kenya compared to the rural population[53]. For instance, urban areas have a higher prevalence of obesity, hypertension, diabetes, and elevated cholesterol compared to rural areas[54]. A previous study in Kenya found that urban residents spent less time in performing moderate to vigorous intensity physical activity and had significantly higher sedentary times compared to those residing in rural areas[53]. Urbanisation also comes with a nutritional transition from traditional fresh foods to western-style processed foods high in fat, salt and sugar alongside the consumption of sugar sweetened beverages[55–58]. Previous studies in Kenya have also found an association between rural-urban migration and risk for overweight and obesity[59], with almost half of urban residents not consuming the minimum required diversity in their diets[60]. Kenya has a high urbanisation rate of 4.23% and currently has about 27% of the population residing in urban areas[61]. With projections that more than half of the Kenyan population will reside in urban areas by 2050[61], effective policies are required to reduce exposure to CVD risk factors that would impact overall CVH status. Moreover, health promotion and prevention interventions targeted at improving CVH should be scaled up in urban areas. For instance, urban designers and planners need to consider increasing provisions for physical activity to reduce sedentary behaviour. The mechanisms driving the urban-rural inequalities in CVH status require further investigation.

Socioeconomic factors, such as wealth and education, accounted for 39.2% of the observed inequalities in iCVH, with wealth contributing 30.7%

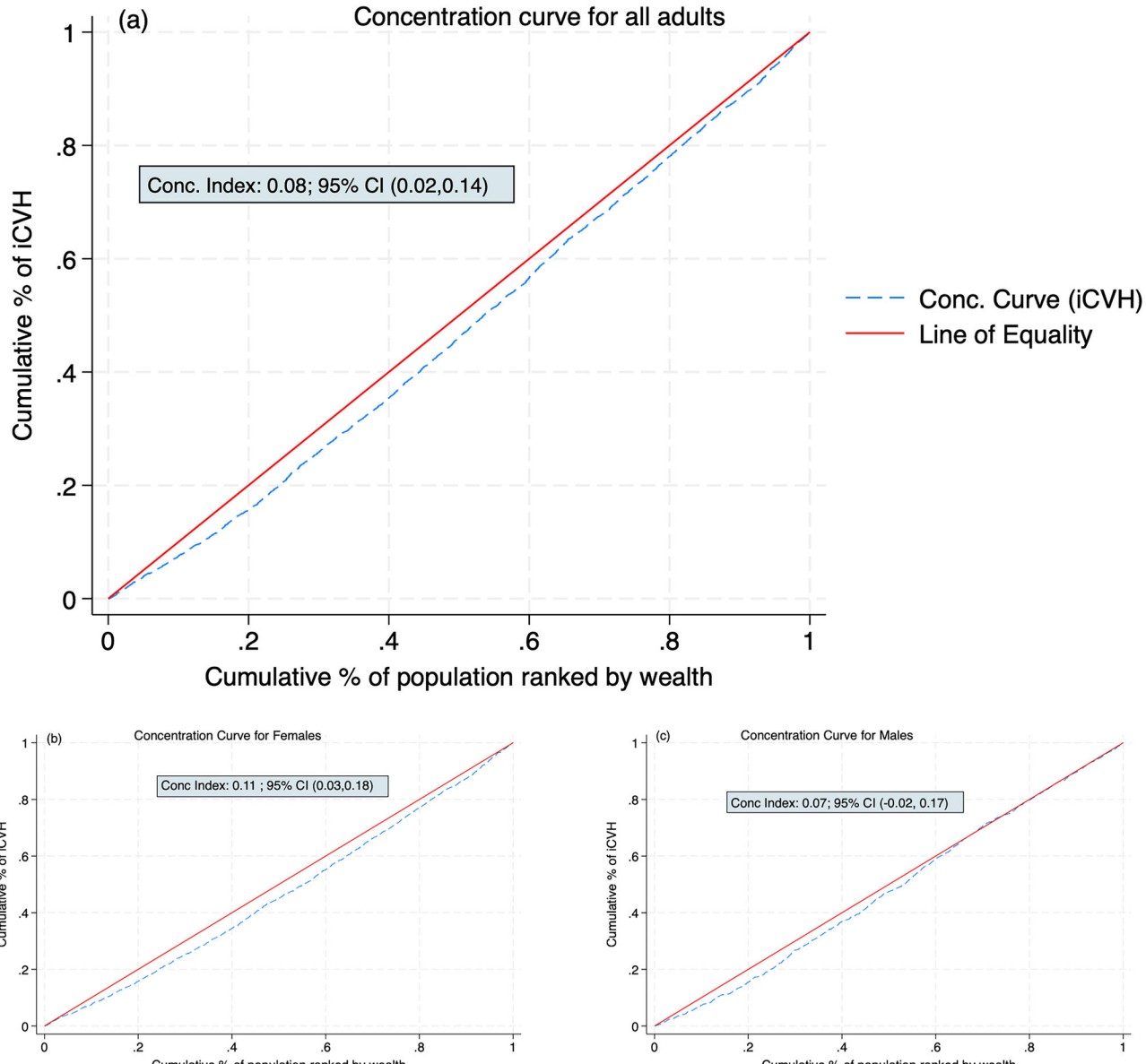

**Fig. 4 | Concentration curve for the wealth-based inequalities in ideal cardiovascular health. a** A graph presenting the concentration curve for the wealth-based inequalities in ideal cardiovascular health among all adults ($n = 3816$). The curve lies below the line of equality indicating a pro-rich distribution of ideal cardiovascular health within the population. The corresponding normalised concentration index is positive and statistically significant confirming the pro-rich inequality. **b** A graph presenting the concentration curve for the wealth-based inequalities in ideal cardiovascular health among females ($n = 2248$). The curve lies below the line of equality indicating a pro-rich distribution of ideal cardiovascular health among females. The normalised concentration index is positive and statistically significant confirming the pro-rich inequality. **c** A graph presenting the concentration curve for the wealth-based inequalities in ideal cardiovascular health among males ($n = 1568$). The curve lies below, but crosses, the line of equality, indicating non-significant pro-rich distribution of ideal cardiovascular health among males. The normalised concentration index is positive, but not statistically significant. Abbreviations (**a–c**): iCVH Ideal cardiovascular health, Conc Index Concentration Index, CI Confidence Interval.

and education 8.5%. The share of inequality explained by wealth was entirely attributed to the richer and richest wealth quintiles while almost all of education's contribution was attributed to those who had secondary or higher education. These findings are consistent with those of previous studies in Kenya[19,20] and Tunisia[46] that found wealth and education to be significant contributors to the observed inequalities in hypertension[19], tobacco use[20]. and high CVD risk[46]. The people living in poverty in Kenya are also likely to be less educated hence not sensitised on the benefits of healthy living, which may increase their CVD risk. Furthermore, the urban poor are likely to consume unhealthy and calorie dense foods that may increase their risk of being overweight and obese[62]. The link between educational attainment and CVD risk has been established, with highly educated individuals more likely to be at low risk for CVDs[63]. In Kenya, a previous study estimated that about 39% of CVD deaths could be averted if individuals attained at least primary education[64]. Highly educated individuals are more likely to practise healthier lifestyles leading to reduced CVD risk factors. The fact that wealth and education are significant contributors to the observed inequality justify continued investment in educational and poverty reduction interventions to promote literacy levels and reduce income inequalities that determine the distribution of CVH status of the population. It is noteworthy that education was an important contributor to the observed inequality among females but not males. While Kenya has made significant strides towards addressing the female disadvantage in education[65,66], our finding highlights the role that the gap in educational

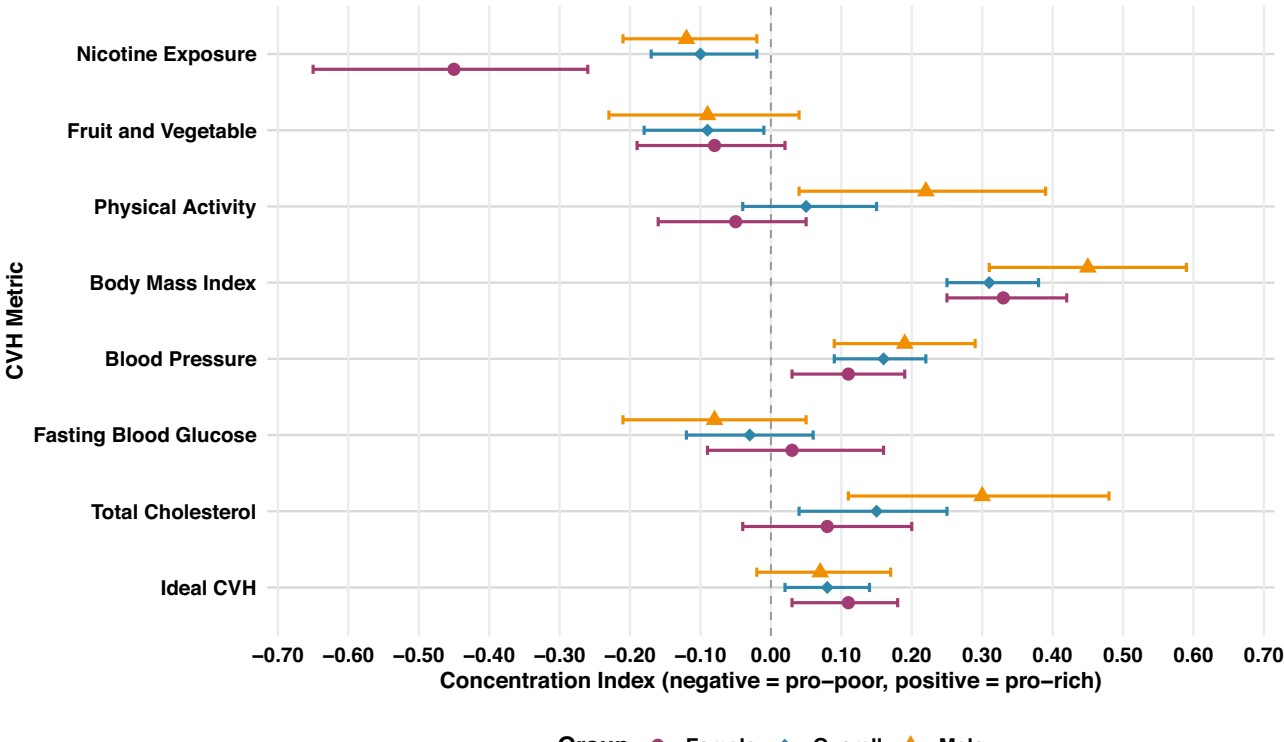

**Fig. 5 | Concentration indices for the wealth-based inequality in ideal cardio-vascular health metrics.** A graph presenting the concentration indices for the wealth-based inequality in ideal cardiovascular health metrics among all adults (overall) and by sex (males and females). The total sample was 3816 adults (2248 females and 1568 males). CVH- Cardiovascular Health; Error bars represent the 95% Confidence Interval. The error bars that do not cross zero are interpreted as statistically significant

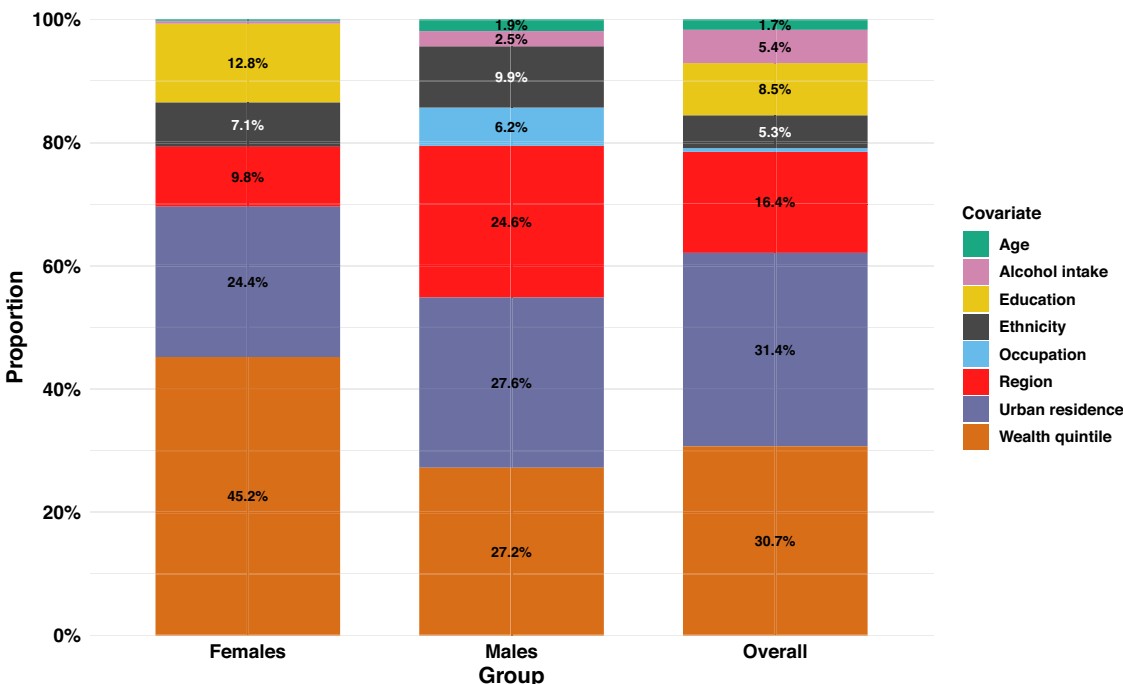

**Fig. 6 | Wagstaff-type decomposition analysis results for the contribution of covariates to the Concentration Index.** A graph presenting the adjusted contributions of the covariates to the inequality among all adults (overall), and by sex (males and females). The total sample was 3816 adults (2248 females and 1568 males). All the analyses are weighted. Each stacked bar represents a population group. Each covariate is represented by a unique colour.

attainment among females plays in influencing the distribution of their CVH status. This finding further points at the need for implementing cardiovascular health education and promotion interventions targeting the least educated women.

Region of residence contributed about 16.4% of the observed inequalities in iCVH, with Nairobi and Central regions being the main drivers. Nairobi is the capital city of Kenya and neighbours the Central region, which also has many urban areas. Nairobi and Central regions have the highest prevalence of obesity[67] and hypertension[19,22]. Furthermore, the Central region is predominantly inhabited by the Kikuyu ethnic group who have been found to have a higher prevalence of CVD risk factors compared to other ethnic groups in Kenya[6,54]. In our study, ethnicity contributed 5.3% of the observed inequality, with almost half (2.4% of total observed inequality) of this contribution attributed to the Kikuyu ethnic group. The mechanisms driving ethnic inequalities in iCVH in Kenya warrant further investigation to inform the design of targeted interventions. Alcohol intake also contributed about 5% of the observed inequality. Individuals who consume alcohol are likely to be predisposed to other risks for CVDs, which can explain its contribution to the observed inequality.

The main strength of this study lies in the use of multiple risk factors to assess the inequalities in the overall CVH status at the population level. We used a nationally representative survey dataset making our findings generalisable to Kenya. Moreover, we performed robust sensitivity analyses by examining inequalities in both ideal and poor CVH, stratified our analyses by sex, and performed further analyses on the imputed dataset. We also decomposed the observed inequalities to identify the drivers of the observed inequalities. Nevertheless, our study has some limitations. The variables included in our model explained only 72.9–78.8% of the observed inequality, leaving over 20% of the variation unexplained, likely due to other factors. Additionally, due to the cross-sectional nature of the data, we cannot infer causation. This analysis is based on the 2015 STEPS survey dataset, which is the first and only available nationally representative survey on NCD risk factors in Kenya. A new round of the STEPS survey would be useful towards monitoring the trends in ideal cardiovascular health over the last decade. Lastly, we used fruit and vegetable intake as a proxy for dietary assessment, which differs from the AHA definition, as we lacked systematically collected data to fully operationalise the diet metric. The study findings are relevant to policy makers regarding the magnitude, patterns and determinants of socioeconomic inequalities in population CVH in Kenya.

## Conclusion

This study finds the existence of pro-rich inequalities in ideal cardiovascular health in Kenya and simultaneous concentration of poor CVH among people living in poverty. Similar pro-rich inequalities were observed in ideal cholesterol and physical activity metrics among males. On the contrary, propoor inequalities were observed in the distribution of ideal smoking metric in all the study groups and ideal FV intake in the combined study sample. Household wealth status, urban residence, region of residence and respondent's education level (mainly among females) were the highest contributors to the observed wealth-based inequalities in iCVH across all the study groups. The study findings highlight the need to adopt equity and gender lens in the design of health promotion and preventive interventions targeting CVD risk factors in Kenya. It is imperative that policy makers in Kenya address the socioeconomic and geographical determinants of health in order to reduce the inequalities in iCVH in Kenya.

## Data availability

This study used publicly available dataset from the Kenya National Bureaus of Statistics (KNBS) and can be requested from the KNBS portal[68]. The source data underlying the figures are contained both in the supplementary tables and from this GitHub repository[27]. Specifically, the source data underlying Figs. 5 and 6 are contained in Supplementary Table 1 and Supplementary Table 5, respectively. Moreover, the source data for reproducing Figs. 1, 2, 3, 5 and 6 are contained in the GitHub repository[27]. Figure 4

was generated based on the entire dataset, which can be accessed from the KNBS portal[68].

## Code availability

The code used in this study is stored in this GitHub repository[27].

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

## Acknowledgements
The authors thank the Kenya National Bureau of Statistics for granting access to the dataset used in this study. This work was funded by the Wellcome Trust as part of a doctoral training grant [218462/Z/19/Z] awarded to J.O.O. to pursue PhD in Public Health Economics and Decision Science at the University of Sheffield. The funders had no role in the design, analyses or reporting of the findings of this study.

## Author contributions
Concept and design: J.O.O. Acquisition of data: J.O.O. Analysis and interpretation of data: J.O.O with reviews from E.W., C.A., P.B., and P.D. Drafting of the manuscript: J.O.O. Critical revision of the paper for important intellectual content: J.O.O., G.M., E.W., C.A., P.O., G.G., E.O., Y.K., O.O., P.B., and P.J.D. Obtaining funding: P.B. and P.J.D. Administrative, technical, or logistic support: J.O.O., P.B., and P.J.D. Supervision: P.B. and P.J.D.

## Competing interests
The authors declare no competing interests.
