## [Transparent Peer Review file · Communications Medicine]

Decomposing Wealth-based Inequalities in Ideal Cardiovascular Health in Kenya

Corresponding Author: Mr James Oguta

Version 0:

Reviewer comments:

Reviewer #1

(Remarks to the Author)

I enjoyed reading this paper by Oguta et al. on the socioeconomic inequalities in ideal cardiovascular health in Kenya. The study was appropriately conducted and is well reported. I have few minor comments:

- 1- You said that the study aimed to assess gender differences in ideal CVH inequalities. However, in some tables you are talking about sex i(male and female). Please use the appropriate terminology about gender/sex and be consistent.
- 2- The Results section is too long. Be concise. Only the most salient findings should be presented in the text. You should avoid duplicating in the text what is already presented in tables.
- 3- In Table 2, the sample for Nairobi is only 51. Is it too small to be representative of the population of this region?
- 4- Underweight has been associated with increased risk of stroke, myocardial infarction, and mortality (PMID: 33619889). Defining ideal BMI as below 25 kg/m² does not account for the risk of cardiovascular disease associated with underweight. This should be discussed.
- 5- You should write "AHA 2010 guidelines" with "s" at the end of "guideline".

Reviewer #2

(Remarks to the Author)

The authors examined socioeconomic inequalities in ideal cardiovascular health in Kenya using 2015 population-based survey data and Wagstaff-type decomposition methods. While the topic is important, and statistical methods appear robust, I have several substantive concerns about the manuscript as submitted.

Major Concerns:

1. Limited Novel Contribution

The knowledge contribution is not sufficiently articulated in this manuscript. The authors note that prior studies have been conducted, and their own data are from 2015, yet they do not adequately justify why this analysis provides new insights for the region or the field. Furthermore, the authors use 2010 AHA recommendations without explaining why more recent guidelines were not considered.

2. Conceptual Issues

Socioeconomic status is widely conceptualized as a multidimensional construct encompassing both wealth and education. However, the authors classify education as an "other covariate" while measuring socioeconomic position solely through an asset index. The lack of conceptual clarity regarding socioeconomic position further limits any potential contribution of this paper. This approach requires justification through appropriate citations and should employ consistent, conventional terminology throughout.

3. Results Presentation and Analysis

The tables are overly dense with information and should be simplified to include only key findings. Auxiliary information not central to the objective should be moved to supplemental materials.

Table 4 appears redundant, essentially mirroring Table 3. The authors should focus on either ideal or poor health metrics,

presenting the other in supplementary materials.

In Table 5, the inclusion of wealth as a covariate in a decomposition of wealth-defined inequalities presents circular logic. Additionally, factors such as region and ethnicity, while potentially important, lack generalizability for an international audience. Reported fixed effects would be better suited for supplementary materials.

Table 3 reveals a critical distinction that lacks discussion in the abstract: the overall ideal CVH score masks diverging findings between health behaviors (smoking concentrated among the poor) and metabolic factors (blood pressure and cholesterol concentrated among the wealthy). This important pattern was discussed in the methods and results, but is omitted from the abstract, which is usually the “take home” for the average reader.

Minor Comments:

Approximately 18% of the original sample was excluded due to missing data. These exclusions require further explanation, and the potential for bias should be evaluated and reported.

The manuscript shows inconsistent statistical reporting, with p-values shown for overall sample comparisons but not by gender in Table 1. This approach should be standardized throughout.

The figures use variable names rather than proper variable labels, affecting readability.

The manuscript requires consistent use of past tense throughout, including the introduction. Multiple proofing errors need attention, such as on page 7, lines 166-7: "We performed the normalisation of the C was performed both at the point of computing the C and during decomposition analysis."

Non-standard acronyms, such as "C" for concentration curve, should be spelled out to improve readability.

The study design aligns more appropriately with "behavioral and social sciences" rather than "life sciences."

Version 1:

Reviewer comments:

Reviewer #1

(Remarks to the Author)

The authors have appropriately addressed the concerns raised in my previous report.

Reviewer #2

(Remarks to the Author)

I appreciate the thorough responses to prior comments. I especially commend the application of multiple imputation. A few remaining issues should be resolved to strengthen the paper:

1) A focus on wealth-based inequalities is valuable. “Wealth” should not be labeled as “socioeconomic” however. The paper title will be clearer if “wealth” is used instead of “socioeconomic.” This is especially important given that you do have measures of education in the analysis which you are not using in your socioeconomic definition.

2) Table 2 CVH metrics in the first column should be more clearly labeled. Are these “ideal” measures (and if so, list thresholds), or the continuous measure as implied by the label – e.g., is this BMI or BMI<25? This will ease the burden in interpreting the meaning of pro-rich/pro-poor inequalities. I will say that the results are surprising in comparison to other LMIC contexts – I would have expected ideal nicotine exposure to be more common in the wealthier groups, and ideal BMI and BP levels to be worse in the wealthier groups.

3) Please use consistent decimal places in Table 2 (a few instances where the 0 is removed)

4) The rationale for including wealth in the Table 3 analysis is fine. I would move the socioeconomic explanatory factors – wealth, education, and occupation – to the top of the table, to give the description more coherence and focus.

5) Table 3 would be much more impactful if the specific levels of each domain were not reported but rather their contribution to inequality (the bolded numbers). The rest could be reported in an appendix. I noted that both reviewers concurred on the enhanced impact of simplifying the presentation.

6) The most obvious evidence of the wealth inequalities in iCVH and other metrics would be a bar graph of the outcomes by wealth quantile. For example, see figure 1 of this article: <https://bmcmmedicine.biomedcentral.com/articles/10.1186/s12916-025-04032-y#Sec1>

This is not mandatory but would substantially enhance the impact of the findings. The current concentration curve figures are not a powerful demonstration of the posited inequality. I appreciate that they show something different, so am suggesting simple bar charts as an addition rather than replacement.

RESPONSE TO REVIEWERS

Reviewer 1:

Referee expertise:

Referee #1: Health equity; CVD in Africa; physician

Referee #2: Epidemiology; decomposition methods; global health

Reviewers' comments:

Reviewer #1 (Remarks to the Author):

I enjoyed reading this paper by Oguta et al. on the socioeconomic inequalities in ideal cardiovascular health in Kenya. The study was appropriately conducted and is well reported.

Response: We appreciate the reviewer's feedback regarding the readability of the paper and the appropriateness of the analyses and its reporting.

I have few minor comments:

1- You said that the study aimed to assess gender differences in ideal CVH inequalities. However, in some tables you are talking about sex i(male and female). Please use the appropriate terminology about gender/sex and be consistent.

Response: We thank the reviewer for this comment and pointing out the inconsistent use of the terms gender and sex. We have now reviewed the entire manuscript and ensured the consistent and appropriate use of the term "sex" meaning the biological construct, which is most relevant to our analysis.

2- The Results section is too long. Be concise. Only the most salient findings should be presented in the text. You should avoid duplicating in the text what is already presented in tables.

Response: Thank you for this comment. We have revised the results section to make it more concise. We have substantially shortened the results section by summarizing key findings and

referring the readers to the relevant tables for additional details. We have removed redundant numeric descriptions. This has reduced the results word count from 1332 words to 619 words.

3- In Table 2, the sample for Nairobi is only 51. Is it too small to be representative of the population of this region?

Response: We thank the reviewer for this comment. The multistage cluster sampling design employed by the STEPS survey ensured that the individuals selected from each county/region represented the population of the region. In our analysis, we accounted for the clustered sampling design of the survey by accounting for the primary sampling unit and strata using the svyset command in STATA, making our analyses nationally representative. All our models attained convergence and did not result in any wide confidence intervals to suggest the lack of statistical power attributed to low sample size. Below is an example of a binary logistic regression model for the odds of having an ideal cardiovascular health by region. The unweighted analysis (n=3816) shows that residing in the Nairobi region is not statistically significant, but the weighted results (Weighted N= 14,639,274) show that residents of Nairobi have reduced odds (statistically significant) of having ideal cardiovascular health. The confidence interval for the odds ratio is also narrow - 0.71[0.55-0.91]

Figure 1: Unweighted analysis of the association between region and ideal cardiovascular health

```
. logistic ideal_final i.region
```

Logistic regression

Log likelihood = -2624.8397

Number of obs = 3,816
 LR chi2(7) = 35.28
 Prob > chi2 = 0.0000
 Pseudo R2 = 0.0067

ideal_final	Odds ratio	Std. err.	z	P> z	[95% conf. interval]
region					
Eastern	.7250433	.0696261	-3.35	0.001	.6006519 .8751953
Nyanza Coast	1.138403	.1229329	1.20	0.230	.9212492 1.406743
Nairobi	.6566069	.1906681	-1.45	0.147	.3716466 1.160061
Western	.9639217	.115154	-0.31	0.758	.7627 1.218231
North Eastern	.970355	.1563922	-0.19	0.852	.7075259 1.330819
Central	.6764134	.0767644	-3.44	0.001	.5415164 .8449146
_cons	1.066087	.0618923	1.10	0.270	.9514273 1.194565

Note: _cons estimates baseline odds.

Figure 2: Weighted analysis of the association between region and ideal cardiovascular health

```
. svy: logistic ideal_final i.region
(running logistic on estimation sample)
```

Survey: Logistic regression

```
Number of strata = 2
Number of PSUs = 200
Number of obs = 3,816
Population size = 14,639,274
Design df = 198
F(7, 192) = 3.11
Prob > F = 0.0039
```

ideal_final	Linearized		t	P> t	[95% conf. interval]	
	Odds ratio	std. err.				
region						
Eastern	.8003591	.1227695	-1.45	0.148	.5914439	1.083069
Nyanza	1.190002	.2298035	0.90	0.369	.8131293	1.741549
Coast	.6388656	.1286984	-2.22	0.027	.4294196	.9504671
Nairobi	.708199	.0891878	-2.74	0.007	.5524577	.9078447
Western	1.024333	.1482419	0.17	0.868	.7700115	1.362652
North Eastern	1.093326	.2837626	0.34	0.731	.655344	1.82402
Central	.540731	.1195349	-2.78	0.006	.3496681	.8361929
_cons	1.260606	.1072984	2.72	0.007	1.065817	1.490996

Note: **_cons** estimates baseline odds.

4- Underweight has been associated with increased risk of stroke, myocardial infarction, and mortality (PMID: 33619889). Defining ideal BMI as below 25 kg/m² does not account for the risk of cardiovascular disease associated with underweight. This should be discussed.

Response: Many thanks for pointing this out and we acknowledge existing evidence associating underweight with increased risk for cardiovascular diseases ¹. We have added this emerging evidence to the discussion section by including this statement:

“

5- You should write "AHA 2010 guidelines" with "s" at the end of "guideline".

Response: Many thanks for this observation. We have made corrections throughout the manuscript to read, “AHA 2010 guidelines”.

Reviewer 2:

Reviewer #2 (Remarks to the Author):

The authors examined socioeconomic inequalities in ideal cardiovascular health in Kenya using 2015 population-based survey data and Wagstaff-type decomposition methods. While the topic is important, and statistical methods appear robust, I have several substantive concerns about the manuscript as submitted.

Major Concerns:

1. Limited Novel Contribution

The knowledge contribution is not sufficiently articulated in this manuscript. The authors note that prior studies have been conducted, and their own data are from 2015, yet they do not adequately justify why this analysis provides new insights for the region or the field.

Response: We thank the reviewer for this comment. To the best of our knowledge, this is the first study to examine the socioeconomic inequalities in ideal cardiovascular health in SSA. While previous studies have only assessed the prevalence and factors associated with individual CVD risk factors, our paper is, to our knowledge, the first to employ Wagstaff-type decomposition methods to comprehensively quantify and explain socioeconomic inequalities in ideal cardiovascular health in Kenya. This methodological approach provides insights into the drivers of these inequalities, which is important and new for the SSA region. The 2015 STEPs dataset is the first and latest available nationally representative dataset available for use by policy makers in Kenya and thus is the most relevant for this analysis. We also emphasize that despite the 2015 data, the persistent nature of health inequalities in low- and middle-income countries makes these findings highly relevant for current policy and intervention design. For instance, Kenya's Gini index has almost remained constant, ranging 39.1% in 2015, to 40.9% in 2019, 38.9% in 2021 and 38.4% in 2022 ².

We have strengthened the introduction and discussion sections to further highlight the novelty. See Lines 73-82 and Lines 289-300. We have also updated the limitations section of the manuscript to highlight the need for new data collection to examine the changing trends.

Furthermore, the authors use 2010 AHA recommendations without explaining why more recent guidelines were not considered.

Response: Thank you for this comment. We acknowledge that the 2010 AHA guidelines ³ were updated in 2022 to assess overall CVH status on an ordinal scale ranging from 0-100, which is

constructed from a simple average of scores from eight CVH metrics (the seven included in our study and sleep health, which was introduced in 2022). However, the 2022 guidelines⁴ did not have a clear operational definition of ideal CVH and ideal levels of each CVH metric. We used the 2010 AHA guideline because it explicitly states the criteria for defining ideal CVH and ideal levels of each of the CVD risk factors (metrics), which is necessary for the design of this study. We have added this to the methods section of our study (Line 113-116).

Table 1: Operational definition of ideal cardiovascular health

Metric	Poor	Intermediate	Ideal
Body mass index	$\geq 30 \text{ kg/m}^2$	25-29.9 kg/m^2	Below 25 kg/m^2
Smoking	Currently smoking or using smokeless tobacco	Past smoker or user of smokeless tobacco ≤ 12 months	Has never smoked or quit >12 months
Fruit and vegetable intake	1 serving or none	2-4 servings	Five or more servings
Physical activity \geq	<600 metabolic equivalent of task (MET) minutes per week	600-1500 metabolic equivalent of task (MET) minutes per week	>1500 metabolic equivalent of task (MET) minutes per week
Blood pressure (BP)	Systolic BP (SBP) $\geq 140\text{mmHg}$ or diastolic BP (DBP) $\geq 90\text{mmHg}$ or existing hypertension patient	SBP $120 \leq 139\text{mmHg}$ or DBP $80 \leq 89\text{mmHg}$	SBP $<120\text{mmHg}$ or DBP $<80\text{mmHg}$ without antihypertensives
Blood Sugar	Fasting blood glucose (FBG) $\geq 7 \text{ mmol/l}$ ($\geq 126 \text{ mg/dl}$) or existing diabetic patient	FBG of 5.6-6.9 mmol/l (100-125 mg/dl)	FBG of $<5.6\text{mmol/l}$ ($<100 \text{ mg/dl}$) without treatment

Total cholesterol (TC)	TC $\geq 6.3\text{mmol/l}$ ($\geq 240\text{ mg/dl}$)	of TC of 5.2-6.2mmol/l (200-239 mg/dl)	<5.2mmol/l (200 mg/dl)
CVH Status	0-2 ideal metrics	3-4 ideal metrics	5-7 ideal metrics

Key: CVH- Cardiovascular Health

2. Conceptual Issues

Socioeconomic status is widely conceptualized as a multidimensional construct encompassing both wealth and education. However, the authors classify education as an "other covariate" while measuring socioeconomic position solely through an asset index. The lack of conceptual clarity regarding socioeconomic position further limits any potential contribution of this paper. This approach requires justification through appropriate citations and should employ consistent, conventional terminology throughout.

Response: Thank you for this comment. We agree that socioeconomic status has been widely conceptualized as a multidimensional construct encompassing both education and wealth. The multidimensional construct is particularly applicable in high income countries where data on income levels is easily available and educational attainment can be quantified in terms of years of school attended. In low and middle income countries, it is particularly difficult to identify income levels due to lack of income data. Moreover, an individual's or household wealth may not necessarily be related to their education status due to high levels of informal employment. Due to these limitations in LMICs, the World Bank proposed the use of household asset ownership as a proxy (indirect) measure of household wealth, which involves the construction of asset-based wealth indices using methods like principal component analysis (PCA)^{5,6}. A similar approach is widely used and acceptable in demographic and health survey (DHS) type of studies in LMICs, which is similar to the STEPs survey used in this study.

In this study, we used PCA to construct household wealth index based on a range of household assets, which included type of toilet facility, type of house (floor, walling and roofing materials), cooking fuel, source of drinking water, availability of electricity, ownership of electronics (radio, television, refrigerator, washing machine, phones, computers), assets for transportation (bicycle, motorcycle, car or truck, animal drawn cart), livestock ownership, ownership of agricultural land and employment of a house help. We have explained how we measured household socioeconomic position in the methods section of the manuscript and have added relevant citations to justify the methods used (See Lines 134-145).

3. Results Presentation and Analysis

The tables are overly dense with information and should be simplified to include only key findings. Auxiliary information not central to the objective should be moved to supplemental materials.

Response: Many thanks for this comment. We have revised the results section and tables and sent all the auxiliary information (including sample characteristics, results of prevalence by sociodemographic characteristics, and imputed analyses) to the supplemental materials. We could not reduce the size of the main decomposition table (Table 3) because it provides a picture of the contribution of various covariates to the inequality. For transparency, we have kept the elasticity, concentration index and absolute contribution columns to promote face validity and enable the reader to verify the computations.

Table 4 appears redundant, essentially mirroring Table 3. The authors should focus on either ideal or poor health metrics, presenting the other in supplementary materials.

Response: Many thanks for this comment. We have now presented ideal CVH metrics only in the main text (Table 2) and sent the table with poor CVH metrics to the supplementary materials.

In Table 5, the inclusion of wealth as a covariate in a decomposition of wealth-defined inequalities presents circular logic.

Response: We thank the reviewer for this perceptive comment that needs clarification. The inclusion of wealth quintile as a covariate in the Wagstaff-type decomposition analysis of the concentration index of a health outcome does not constitute a circular logic because, while the concentration index is computed with respect to the ranking of individuals by wealth, the decomposition framework assesses the extent to which observed inequalities in the health variable are explained by covariates, including wealth itself. In other words, wealth ranking is used in the construction of the inequality-focused response variable, but this inequality could be driven by various explanatory factors, including wealth.

This approach is standard in health equity research⁶, where wealth is treated both as a ranking variable and as an explanatory determinant of health. Including it allows us to quantify the degree to which disparities in wealth—beyond its use for ranking—contribute to inequality in the health outcome. Wealth/income variable has been included as a covariate in all the previous studies that have decomposed the socioeconomic inequalities in CVD risk factors⁷⁻¹⁴.

Additionally, factors such as region and ethnicity, while potentially important, lack generalizability for an international audience. Reported fixed effects would be better suited for supplementary materials.

Response: Thank you for the comment. We agree that religion and ethnicity may not vary to other settings, but ours is primarily an analysis of Kenyan data designed to inform Kenyan policy. Region and ethnicity have been established as important determinants of NCD distribution in the Kenyan context ^{7,15,16}. We therefore feel it is important to include these variables within the decomposition analysis and to report their relative contribution to the observed inequalities and inform the design of targeted interventions to improve CVH status in Kenya.

Table 3 reveals a critical distinction that lacks discussion in the abstract: the overall ideal CVH score masks diverging findings between health behaviors (smoking concentrated among the poor) and metabolic factors (blood pressure and cholesterol concentrated among the wealthy). This important pattern was discussed in the methods and results, but is omitted from the abstract, which is usually the “take home” for the average reader.

Response: Many thanks for this comment. We have revised the abstract to highlight the observed patterns across the individual CVH metrics.

Minor Comments:

Approximately 18% of the original sample was excluded due to missing data. These exclusions require further explanation, and the potential for bias should be evaluated and reported.

Response: Thank you for this comment. We performed complete case analysis by focusing only on individuals without missing data on the outcome variables and selected covariates (n=3816). We could not tell why some variables had missing observations, but opine that it could be related to non response or missingness during initial data processing. In the complete case analysis, we had to ensure that we kept only observations that had no missing data on all the important variables included in the analysis, which reduced the initial sample from 4500 to 3816.

As a sensitivity analysis, we have conducted additional extensive analyses to handle missing data by performing multiple imputation using chained equations (MICE) using the MICE package in R ¹⁷. We have assessed the pattern of missingness and performed relevant diagnostics, which indicate that the data on key variables were missing at random (Figure 3) hence justifying the use of MICE for imputation. We then performed 20 imputations, which were increased to 80 imputations, after performing convergence diagnostics (Figure 4 and 5). Predictive mean matching was used to impute continuous data, logistic regression used for binary data while polytomous logistic regression was used to impute unordered categorical data ¹⁷. The results of the analysis on the imputed dataset were consistent with those of complete case analysis presented earlier in this manuscript. We have now presented the results of the socioeconomic inequalities in concentration indices for ideal and poor CVH metrics from the

imputed analyses in the supplementary files. We have not explained how we handled missing data (See Lines 205-214).

Figure 3: Assessing the pattern of missingness

Figure 4: Convergence diagnostics after 20 imputations

Figure 5: Convergence diagnostics after 80 imputations

Iteration

Figure 4: Assessing the distribution of the imputed observations

Stripplots of Key Numerical Variables

Stripplots of Key Categorical Variables

Concentration indices for the socioeconomic inequality in ideal cardiovascular health (CVH) metrics (Imputed analyses-n=4500)

CVH Metric	Overall	Female	Male
Nicotine	-0.11 (-0.18,-0.04)**	-0.49 (-0.65,-0.32)***	-0.13 (-0.22,-0.05)**
Fruit and vegetable intake	-0.09 (-0.17,-0.01)*	-0.09 (-0.18,0.01)	-0.07 (-0.2,0.05)
Physical Activity	0.05 (-0.04,0.13)	-0.06 (-0.15,0.02)	0.23 (0.07,0.4)**
Body Mass Index	0.32 (0.25,0.38)***	0.33 (0.25,0.4)***	0.45 (0.32,0.58)***
Blood Pressure	0.16 (0.1,0.22)***	0.09 (0.02,0.17)*	0.21 (0.11,0.3)***
Fasting Blood Glucose	-0.02 (-0.1,0.07)	0.02 (-0.09,0.14)	-0.04 (-0.18,0.1)
Total Cholesterol	0.14 (0.04,0.23)**	0.08 (-0.03,0.19)	0.31 (0.13,0.49)**
Ideal CVH	0.08 (0.03,0.14)**	0.08 (0.01,0.15)*	0.1 (0.01,0.18)*
Bold is statistically significant at p<0.001(***), p<0.01(**), and p<0.05(*)			

Concentration indices for the socioeconomic inequality in poor cardiovascular health (CVH) metrics (Imputed analyses-n=4500)

CVH Metric	Overall	Female	Male
Nicotine	0.21 (0.12,0.31)***	0.64 (0.55,0.74)***	0.2 (0.1,0.31)***
Fruit and vegetable intake	0.17 (0.11,0.22)***	0.17 (0.1,0.24)***	0.18 (0.09,0.26)***
Physical Activity	0.01 (-0.13,0.15)	0.12 (-0.01,0.24)	-0.17 (-0.4,0.07)
Body Mass Index	-0.34 (-0.41,-0.26)***	-0.33 (-0.41,-0.25)***	-0.55 (-0.69,-0.4)***
Blood Pressure	-0.07 (-0.13,-0.01)*	-0.09 (-0.16,-0.01)*	-0.07 (-0.17,0.04)
Fasting Blood Glucose	-0.12 (-0.29,0.04)	-0.15 (-0.33,0.03)	-0.12 (-0.41,0.17)
Total Cholesterol	-0.22 (-0.39,-0.06)**	-0.14 (-0.32,0.04)	-0.48 (-0.75,-0.22)***
Poor CVH	-0.06 (-0.15,0.03)	-0.04 (-0.14,0.06)	-0.12 (-0.28,0.05)
Bold is statistically significant at $p < 0.001$ (***), $p < 0.01$ (**), and $p < 0.05$ (*)			

The manuscript shows inconsistent statistical reporting, with p-values shown for overall sample comparisons but not by gender in Table 1. This approach should be standardized throughout.

Response: Thank you for this observation. We have now standardized the presentation of statistical significance throughout the manuscript, including Table 2.

The figures use variable names rather than proper variable labels, affecting readability.

Response: Thank you for pointing this out. We have updated all figures to use clear, descriptive variable labels instead of technical variable names, significantly improving their readability and comprehension.

The manuscript requires consistent use of past tense throughout, including the introduction. Multiple proofing errors need attention, such as on page 7, lines 166-7: "We performed the normalisation of the C was performed both at the point of computing the C and during decomposition analysis."

Response: Thank you for the comment. We have thoroughly proofread the entire manuscript to ensure consistent use of past tense throughout the paper.

Non-standard acronyms, such as "C" for concentration curve, should be spelled out to improve readability.

Response: Thank you for the comment. We have now added the definition of all the acronyms used in the abbreviations section.

The study design aligns more appropriately with "behavioral and social sciences" rather than "life sciences."

Response: Thank you for the comment. We appreciate this insight and would be happy to work with the editorial team to ensure proper categorization, if needed.

References

1. Kwon, H. *et al.* Incidence of cardiovascular disease and mortality in underweight individuals. *J. Cachexia Sarcopenia Muscle* **12**, 331–338 (2021).
2. Kenya National Bureau of Statistics (KNBS). *The Kenya Poverty Report: Based on the 2022 Kenya Continuous Household Survey*. <https://www.knbs.or.ke/wp-content/uploads/2024/10/The-Kenya-Poverty-Report-2022.pdf> (2024).
3. Lloyd-Jones, D. M. *et al.* Defining and setting national goals for cardiovascular health promotion and disease reduction: the American Heart Association’s strategic Impact Goal through 2020 and beyond. *Circulation* **121**, 586–613 (2010).
4. Lloyd-Jones, D. M. *et al.* Life’s Essential 8: Updating and enhancing the American heart association's construct of cardiovascular health: A presidential advisory from the American Heart Association. *Circulation* **146**, e18–e43 (2022).
5. D’Alessio, G. & Toma, I. Measuring wealth in household surveys in low- and middle-income countries. An introduction to the world bank guidelines. *Stat. J. LAOS* 18747655251342644 (2025).
6. O’Donnell *et al.* Analyzing health equity using household survey data : a guide to techniques and their implementation. *World Bank* <https://documents.worldbank.org/en/publication/documents-reports/documentdetail/633931468139502235/analyzing-health-equity-using-household-survey-data-a-guide-to-techniques-and-their-implementation> (2008).
7. Gatimu, S. M. & John, T. W. Socioeconomic inequalities in hypertension in Kenya: a decomposition analysis of 2015 Kenya STEPwise survey on non-communicable diseases risk factors. *Int. J. Equity Health* **19**, 213 (2020).
8. Alaba, O. & Chola, L. Socioeconomic inequalities in adult obesity prevalence in South Africa: a decomposition analysis. *Int. J. Environ. Res. Public Health* **11**, 3387–3406 (2014).

9. Mosquera, P. A., San Sebastian, M., Ivarsson, A. & Gustafsson, P. E. Decomposition of gendered income-related inequalities in multiple biological cardiovascular risk factors in a middle-aged population. *Int. J. Equity Health* **17**, 102 (2018).
10. Saidi, O. *et al.* Explaining income-related inequalities in cardiovascular risk factors in Tunisian adults during the last decade: comparison of sensitivity analysis of logistic regression and Wagstaff decomposition analysis. *Int. J. Equity Health* **18**, 177 (2019).
11. Chisha, Z., Nwosu, C. O. & Ataguba, J. E.-O. Decomposition of socioeconomic inequalities in cigarette smoking: the case of Namibia. *Int. J. Equity Health* **18**, 6 (2019).
12. Umuhoza, S. M. & Ataguba, J. E. Inequalities in health and health risk factors in the Southern African Development Community: evidence from World Health Surveys. *Int. J. Equity Health* **17**, 52 (2018).
13. Nglazi, M. D. & Ataguba, J. E. Socioeconomic inequalities in intergenerational overweight and obesity transmission from mothers to offsprings in South Africa. *SSM Popul. Health* **19**, 101170 (2022).
14. Donfouet, H. P. P., Mohamed, S. F. & Malin, E. Socioeconomic inequality in tobacco use in Kenya: a concentration analysis. *International Journal of Health Economics and Management* **21**, 247–269 (2021).
15. Wekesah, F. M. *et al.* Individual and household level factors associated with presence of multiple non-communicable disease risk factors in Kenyan adults. *BMC Public Health* **18**, 1220 (2018).
16. Oguta, J. O. *et al.* Prevalence and determinants of ideal cardiovascular health in Kenya: A cross-sectional study using data from the 2015 Kenya STEPwise survey. *Glob. Heart* **19**, 79 (2024).
17. Zhang, Z. Multiple imputation with multivariate imputation by chained equation (MICE) package. *Ann. Transl. Med.* **4**, 30 (2016).

RESPONSE TO REVIEWERS

REVIEWER 1

Reviewers' comments:

Referee #1: Health equity; CVD in Africa; physician

Reviewer #1 (Remarks to the Author):

The authors have appropriately addressed the concerns raised in my previous report.

Response: We thank the reviewer for this positive assessment and appreciate their time and comments that helped improve the previous draft of the manuscript.

REVIEWER 2

Referee #2: Epidemiology; decomposition methods; global health

Reviewer #2 (Remarks to the Author):

I appreciate the thorough responses to prior comments. I especially commend the application of multiple imputation.

Response: We are grateful for the reviewer's positive feedback. We agree that multiple imputation strengthens the robustness of our findings by addressing potential bias from missing data.

A few remaining issues should be resolved to strengthen the paper:

1) A focus on wealth-based inequalities is valuable. "Wealth" should not be labeled as "socioeconomic" however. The paper title will be clearer if "wealth" is used instead of "socioeconomic." This is especially important given that you do have measures of education in the analysis which you are not using in your socioeconomic definition.

Response: We thank the reviewer for this important clarification. We have revised the title to "Wealth-based Inequalities in Ideal Cardiovascular Health in Kenya: A Decomposition Analysis." We have also replaced "socioeconomic" with "wealth" throughout the manuscript.

2) Table 2 CVH metrics in the first column should be more clearly labelled. Are these "ideal" measures (and if so, list thresholds), or the continuous measure as implied by the label – e.g., is this BMI or BMI<25? This will ease the burden in interpreting the meaning of pro-rich/pro-poor inequalities.

Response: We thank the reviewer for this observation. We have replaced the previous Table 2 with Figure 5, which is a forest plot of the concentration indices depicting the inequalities in ideal CVH metrics in Kenya. Here is the new Figure 5:

We have also moved the previous Table 2 to Supplementary Table 1 and revised it to clearly indicate that the measures presented are ideal CVH metrics and have included the specific thresholds for each metric (e.g., BMI <25 kg/m², blood pressure <120/80 mmHg, no current smoking). This improves interpretability of the pro-rich/pro-poor inequality estimates.

Comment: I will say that the results are surprising in comparison to other LMIC contexts – I would have expected ideal nicotine exposure to be more common in the wealthier groups, and ideal BMI and BP levels to be worse in the wealthier groups.

Response: Thank you for this observation. We have compared our results with previous studies within the African context and highlighted the studies that are consistent and those contrast with our findings. For instance, our finding on the pro-rich inequality (positive C) in ideal BP agrees with those of Gatimu and John (1) who found pro-poor inequality (negative C) in hypertension in Kenya, using the STEPwise survey. Nevertheless, our analyses are robust as we performed multiple sensitivity analyses by assessing the inequalities in both ideal and poor CVH, which were consistent

with each other—Inequalities in poor CVH mirrored those of ideal CVH. Furthermore, the results from the imputed analyses were also consistent with those of complete case analyses.

3) Please use consistent decimal places in Table 2 (a few instances where the 0 is removed)

Response: Thanks for this observation. We have now standardized all reported figures in previous Table 2 (now Supplementary Table 1) to two decimal places for consistency.

4) The rationale for including wealth in the Table 3 analysis is fine. I would move the socioeconomic explanatory factors – wealth, education, and occupation – to the top of the table, to give the description more coherence and focus.

Response: Thanks for highlighting this. We have replaced the previous Table 3 with Figure 6 and moved the table to the supplementary files for more details. The revised draft is now visually appealing and has improved coherence. Here is the new Figure 6:

5) Table 3 would be much more impactful if the specific levels of each domain were not reported but rather their contribution to inequality (the bolded numbers). The rest could be reported in an appendix. I noted that both reviewers concurred on the enhanced impact of simplifying the presentation.

Response: We appreciate this helpful suggestion. We have replaced the previous Table 3 with Figure 6, which presents the overall contribution of the covariates to the inequality. We have transferred the previous table 3 to the appendices for further details.

6) The most obvious evidence of the wealth inequalities in iCVH and other metrics would be a bar graph of the outcomes by wealth quintile. For example, see figure 1 of this article:

<https://bmcmedicine.biomedcentral.com/articles/10.1186/s12916-025-04032-y#Sec1>

This is not mandatory but would substantially enhance the impact of the findings. The current concentration curve figures are not a powerful demonstration of the posited inequality. I appreciate that they show something different, so am suggesting simple bar charts as an addition rather than replacement.

Response: We thank the reviewer for this excellent suggestion. We have added new bar charts (Figure 3) depicting prevalence of ideal CVH and component metrics across wealth quintiles. These complement the concentration curves (Figure 4), providing a clearer and more intuitive visualization of inequalities. Here is the new Figure 3:

Reference

1. Gatimu SM, John TW. Socioeconomic inequalities in hypertension in Kenya: a decomposition analysis of 2015 Kenya STEPwise survey on non-communicable diseases risk factors. *Int J Equity Health*. 2020;19(1):213.